# An Unruptured True Aneurysm of the Uterine Artery during Pregnancy

**DOI:** 10.3390/diagnostics12102459

**Published:** 2022-10-11

**Authors:** Charuwan Tantipalakorn, Suchaya Luewan, Sirinart Sirilert, Theera Tongsong

**Affiliations:** Department of Obstetrics and Gynecology, Faculty of Medicine, Chiang Mai University, Chiang Mai 50200, Thailand

**Keywords:** aneurysm, pregnancy, ultrasound, uterine artery

## Abstract

The antenatal diagnosis of an unruptured true aneurysm of the uterine artery is extremely rare and has never been reported, whereas pseudoaneurysms associated with previous trauma or cesarean section have been reported several times. True aneurysms occur when the artery or vessel weakens and bulges, sometimes forming a blood-filled sac. Nearly all cases of pelvic true aneurysms involved ovarian arteries which ruptured during the peripartum period. The case presented here is unique in terms of being an unruptured true aneurysm of the uterine artery with a first diagnosis during pregnancy at 32 weeks of gestation and the spontaneous development of thrombosis in the aneurysm in late pregnancy, documented at 37 weeks of gestation. The diagnosis of a true aneurysm of the uterine artery was based on, (1) a demonstration of the cystic mass located in proximity to the lower segment of the uterus with ultrasound characteristics of arterial flow in the mass, and (2) the occurrence in a woman who had no history of trauma or surgery in the pelvis. The finding during cesarean section confirmed the prenatal sonographic finding. The pregnancy ended with successful outcomes.

**Figure 1 diagnostics-12-02459-f001:**
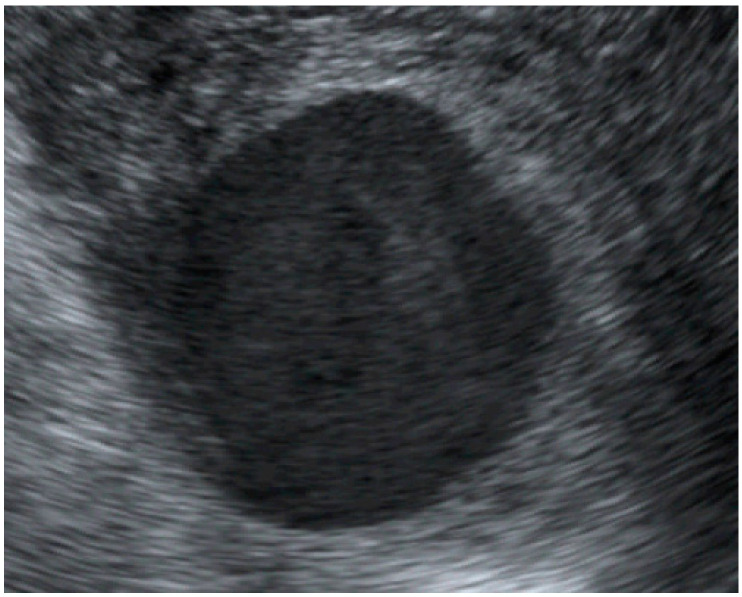
**and Appendix A:** A 32-year-old primigravid woman underwent obstetric ultrasound at 32 weeks of gestation because of a small-for-date uterine size. Her medical history as well as familial history was unremarkable; no underlying medical disease and no history of any pelvic surgery. The antenatal course of the current pregnancy was uneventful. Ultrasound screening for fetal anomaly at 20 weeks of pregnancy revealed normal structures and fetal biometry, with no records of pelvic pathology on sonographic examination. Ultrasound examination at this visit (32 weeks of pregnancy) showed a slightly delayed growth of the fetus (estimated fetal weight of 15th percentile). Interestingly, a cystic mass at the right adnexa, close to the lower uterine segment was noted. The mass was measured as 3.0 × 4.5 × 3.0 cm in size, well-circumscribed, unilocular, and had homogeneous low-level echoes, with swirling flow in the mass, which can be clearly visualized on the simple 2D ultrasound as seen in Figure 1 and Appendix A. A uterine artery aneurysm was highly suspected upon 2D ultrasound. Aneurysms of the uterine artery are rare and with an unknown true prevalence. The entity can be categorized into two groups, pseudoaneurysms and true aneurysms. Pseudoaneurysms are abnormal outpouchings or the dilatation of arteries which are bounded only by the tunica adventitia, the outermost layer of the arterial wall, whereas true aneurysms are bounded by all three layers of the arterial wall. Pseudoaneurysms typically occur when a blood vessel wall is injured and the leaking blood collects in the surrounding tissue. They can occur in patients of all ages, typically following penetrating or blunt trauma, infection, dissection, excessive effort, or as a complication of a cesarean section [1]. True aneurysms occur when the artery or vessel weakens and bulges, sometimes forming a blood-filled sac. True aneurysms of uterine arteries are extremely rare. To the best of our knowledge, a very limited number of isolated case reports have been published in the literature [2,3]. Moreover, a true aneurysm of the uterine artery has never been described during pregnancy. Therefore, its natural history remains unexplored. Nearly all cases reported in the literature are pseudoaneurysms. Additionally, most published pelvic true aneurysms involved ruptured ovarian artery aneurysms during peripartum periods [4,5,6,7,8]. The case presented here is unique in terms of being an unruptured true aneurysm of the uterine artery at the time of diagnosis and with the spontaneous development of thrombosis in the aneurysm in late pregnancy.

**Figure 2 diagnostics-12-02459-f002:**
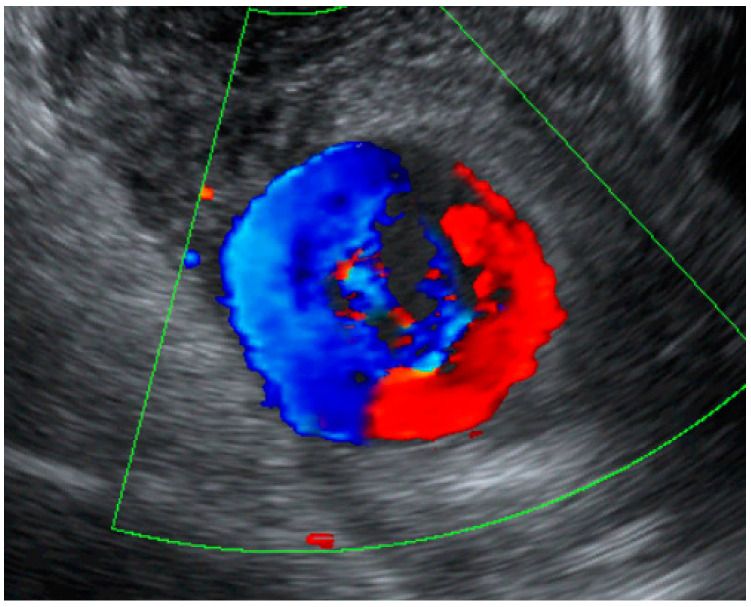
**and Appendix A:** A color flow ultrasound of the same mass as Figure 1 shows swirling flow in the mass, which is consistent with the arterial flow as presented in Figure 2 and Appendix A. A uterine artery aneurysm was diagnosed based on the finding of active swirling blood flow in the saccular-like sac connecting the uterine artery.

**Figure 3 diagnostics-12-02459-f003:**
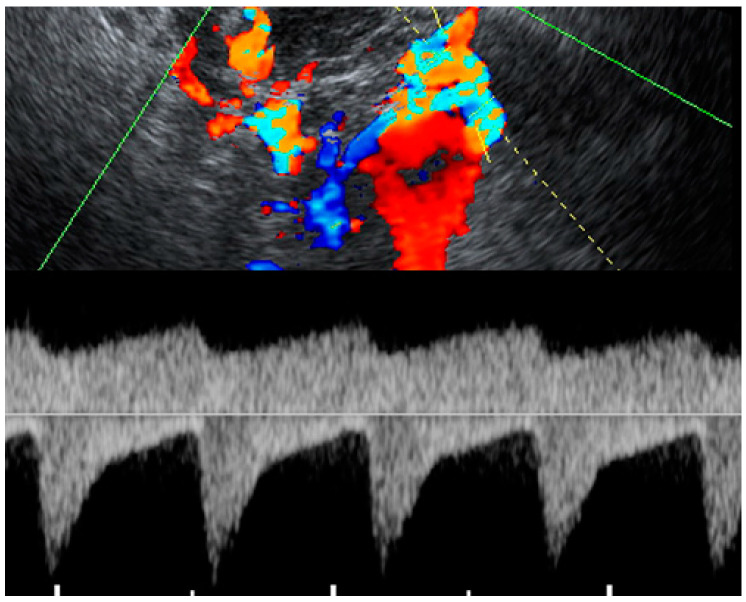
Spectral Doppler of the flow in the same mass as Figure 2 shows a bidirectional flow in the mass. The waveforms below the baseline maintain the typical characteristics of uterine artery waveforms, whereas the waveforms above the baseline swirl back in the mass toward the transducer and lose the typical characteristic waveforms of the uterine artery, as presented in Figure 3. The provisional diagnosis of the mass was a true aneurysm of the uterine artery, based on the location of the mass, ultrasound characteristics and no history of trauma or surgery in the pelvis. The management plan after counseling the couple about the risk of an aneurysm rupture was conservative treatment with close monitoring. A cesarean section was scheduled to avoid compression pressure from the gravid uterus during labor and vaginal delivery which might theoretically facilitate the rupturing of the aneurysm.

**Figure 4 diagnostics-12-02459-f004:**
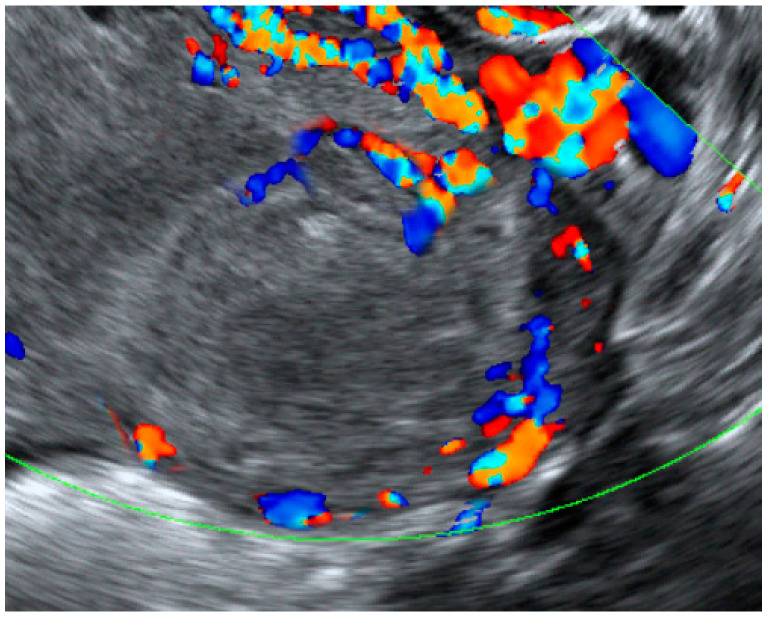
Color flow ultrasound of the same mass at a follow-up ultrasound at 37 weeks of gestation shows changes in the active flow of the cyst, now as an inactive hemorrhagic heterogeneous mass without internal color flow, even at the low PRF setting, as presented in Figure 4. The finding suggested that the aneurysm had become thrombotic. Cesarean section was performed at 39 weeks of gestation because of a breech presentation, giving birth to a healthy female newborn with Apgar scores of 8 and 10 at 1 and 5 min. The operative finding revealed the unruptured thrombotic aneurysm close to the right uterine isthmus and cardinal ligament. The thrombotic uterine artery was approximately 4 cm in length and 1 cm in diameter with a saccular-like area of 2 cm in width connected to the uterus. The patient had an uneventful postpartum period but did not attend a follow-up at 6 weeks postpartum. **Clinical Impact**. An unruptured true aneurysm of the uterine artery can be detected by one of the following techniques: (1) Color Doppler ultrasound finding an intrauterine mass or a mass connected to the uterus with swirling blood flow, with a to-and-fro pattern; (2) magnetic resonance imaging (MRI) revealing an enhanced, sac-like structure within the uterus or connected to the uterus; (3) computed tomography angiography (CTA) confirming the presence of sac-like structure with a connection to the uterine artery. Preoperatively, differentiating a true aneurysm from a pseudoaneurysm may be based on the presence of a prior history of trauma or infection, pelvic surgery or cesarean section. Nevertheless, definite diagnosis relies on either finding intact vascular wall layers in the operative field or pathological discovery. A uterine artery pseudoaneurysm or arteriovenous fistula, probably also a true aneurysm, is usually detected after the rupture of lesions, resulting in a spontaneous massive hemorrhage or after uterine curettage [9]. Pathologically, pseudoaneurysms usually consist of only one layer of loose connective tissue, different from true aneurysms which consist of a complete three-layered wall. Extraluminal swirling blood flow can lead to the enlargement of the pseudoaneurysm, making it susceptible to rupture and serious bleeding. The natural course and prognosis of true aneurysms are not known. However, it should be considered a serious condition and a difficult cesarean section in the category of laceration or organ damage [10], which can cause massive hemorrhage, leading to a life-threatening scenario. Knowing in advance the potential surgical difficulties allows the surgeon to plan the best strategies. Nevertheless, theoretically, true aneurysms may be less susceptible to rupture than pseudoaneurysms since they have a more secure vascular wall because of intact complete vascular wall layers. However, high precaution must be exercised, especially in late pregnancy and early postpartum. It might be life-threatening as seen in true aneurysms of the ovarian arteries, which are normally detected upon rupture in up to 50% of cases during the peripartum period [4,5,6,7,8], probably because of the anatomical changes in the vessel during the pregnancy, facilitating the weakening of the arterial wall. Additionally, the hormonal and hemodynamic changes induced by pregnancy may lead to the development of these aneurysms. The development of the aneurysm in our case might have been induced by the pregnancy, occurring in the second half of pregnancy since it was not documented during the ultrasound examination at mid-pregnancy. However, it was possible that a pre-existing lesion might have been an overlooked anomaly and missed during routine screening. The management of the aneurysm may follow the guidelines for pseudoaneurysms, such as open laparotomy for hysterectomy or ligation of the uterine artery or internal iliac artery, uterine balloon tamponade and laparoscopic surgery for the treatment [11,12] of transarterial embolization [13]. However, proper management during pregnancy is challenging. Uterine artery or hypogastric artery ligation during pregnancy certainly has a higher risk and needs expertise, as well as the consideration of possible adverse effects on the pregnancy and the fetus. Theoretically, uterine artery embolization is also associated with a negative impact on the pregnancy. Thus, our patient preferred a conservative treatment with close monitoring. Interestingly, the case presented here spontaneously developed a thrombosis in the aneurysm and needed no further treatment. The spontaneous thrombosis was described before by Borghese et al. [1], who described a true uterine artery aneurysm incidentally detected in a 39-year-old Caucasian female patient who was asymptomatic and not pregnant. The CTA showed that the aneurysm arose from the right uterine artery, measuring 13 mm in maximal diameter. The patient refused any treatment and the CTA 3 months later showed spontaneous thrombosis of the aneurysm. Accordingly, spontaneous thrombolysis might represent one of the possible natural outcomes and close follow-up with imaging should be performed, especially during pregnancy, as seen in our case. In summary, we described a unique case of a true aneurysm of the uterine artery diagnosed at 32 weeks of gestation by the demonstration of an adnexal cystic mass with swirling flow consistent with the uterine artery, connected with the uterine isthmus. If ruptured, the aneurysm placed the patient at high risk of massive hemorrhage. However, spontaneous thrombosis occurred in late gestation, ending with a successful outcome through expectant management.

## Data Availability

The data of this report are available from the corresponding authors upon request.

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
