# Peer review of "An Unruptured True Aneurysm of the Uterine Artery during Pregnancy"

_diagnostics, 2022, doi:10.3390/diagnostics12102459_

Round 1
Reviewer 1 Report
Dear authors
what a nice case
I've never heard such a case
literature is scarne about therefore I think it is useful to share this experience to let other obestrician aware of this eventuality
I think a condition like that might enter in the category of potentially difficult Caesarean section due to the risk of bleeding in case of isterotomy extension
I think is fear to mention this eventuality
and read and cite a literature review about difficult Caesarean section PMID: 31962259.
best regards
Author Response
Reviewer: 1 (highlighted in blue)
Dear authors
what a nice case
I've never heard such a case
literature is scarne about therefore I think it is useful to share this experience to let other obestrician aware of this eventuality
I think a condition like that might enter in the category of potentially difficult Caesarean section due to the risk of bleeding in case of isterotomy extension
I think is fear to mention this eventuality
and read and cite a literature review about difficult Caesarean section PMID: 31962259.
best regards
Response:
- In revised MS, we have mentioned the concern of difficult cesarean section, as highlighted in blue.
- In revised MS, we cite the article suggested by the reviewer (Ref 10).

Reviewer 2 Report
discussion must be more long
a little paragraph on arterious malformation needs to be inserted to imporve the paper
(
Thrombosed Arteriovenous Malformation of Umbilical Cord
Damiani, G.R., Arezzo, F., Vimercati, A., ...Gaetani, M., Cicinelli, E. , Acquired uterine arteriovenous fistula following dilatation and curettage: an uncommon cause of vaginal bleeding. Radiol Case Rep. 2017 Feb 21;12(2):287-291. doi: 10.1016/j.radcr.2017.01.005. eCollection 2017 Jun.PMID: 28491172 Free PMC article.Author Response
Reviewer: 2 (highlighted in red)
discussion must be more long
Response: The article type (interesting image) needs no long discussion. Therefore, we make a request to keep the discussion as the way it is since we already summarize the main points in the discussion.
a little paragraph on arterious malformation needs to be inserted to imporve the paper
Response: In revised MS, we mentioned on arteriovenous malformation, as highlighted in red.
Thrombosed Arteriovenous Malformation of Umbilical Cord
Damiani, G.R., Arezzo, F., Vimercati, A., ...Gaetani, M., Cicinelli, E. Journal of Obstetrics and Gynecology of Indiathis link is disabled, 2022
Acquired uterine arteriovenous fistula following dilatation and curettage: an uncommon cause of vaginal bleeding.
Evans A, Gazaille RE 3rd, McKenzie R, Musser M, Lemming R, Curry J, Meyers W, Austin N.
Radiol Case Rep. 2017 Feb 21;12(2):287-291. doi: 10.1016/j.radcr.2017.01.005. eCollection 2017 Jun.
PMID: 28491172 Free PMC article. these 2 papers coul help to make more interesting the paper if integrated in the discussion cocnlusion need to be revised
Response: The first article could not found on PubMed. The second one is now cited, as highlighted and added in the “Reference 9”.
